# Prevalence and Haplotypes of *Toxoplasma gondii* in Native Village Chickens and Pigs in Peninsular Malaysia

**DOI:** 10.3390/vetsci10050334

**Published:** 2023-05-06

**Authors:** Sabrina Danial Leong, Latiffah Hassan, Reuben Sunil Kumar Sharma, Ooi Peck Toung, Hassan Ismail Musa

**Affiliations:** 1Faculty of Veterinary Medicine, Universiti Putra Malaysia, 44300 Serdang, Malaysia; sabrina_danial@hotmail.com (S.D.L.); reuben@upm.edu.my (R.S.K.S.); ooi@upm.edu.my (O.P.T.); hassanismail@upm.edu.my (H.I.M.); 2Faculty of Veterinary Medicine, University of Maiduguri, PMB 1069 Maiduguri, Nigeria

**Keywords:** *Toxoplasma gondii*, prevalence, haplotypes, village chicken, pig, risk factor

## Abstract

**Simple Summary:**

*Toxoplasma gondii* is an emerging foodborne parasite disease causing morbidity and mortality worldwide. The coccidian parasite, *Toxoplasma gondii*, has an exceptionally broad spectrum of intermediate hosts. Felids are the only definitive host responsible in producing oocysts that are highly resistant to environmental elements. Toxoplasmosis is common leading to public health concerns and economic losses to the animal industry. In the current study, serum and tissue samples from village chickens and pigs were examined. In chickens, 7.6% of serum and 14.0% of tissue samples were detected with *T. gondii.* On the other hand, 3.0% of serum and 5.8% of pig tissue samples were positive with *T. gondii.* Six unique DNA sequences were isolated from the tissue samples. The risk factor findings, which are the first study in Malaysia, emphasised the necessity for covering feed storage and tightening the biosecurity in farm to lower the exposure of village chickens and pigs to the parasite.

**Abstract:**

*Toxoplasma gondii* is an important zoonotic foodborne parasite capable of infecting almost all warm-blooded animal species worldwide. Toxoplasmosis is usually acquired via ingestion of undercooked infected animal tissues resulting in life-threatening consequences for unborn foetus and immunocompromised individuals. A cross-sectional study was carried out to determine the prevalence of *T. gondii* infection, its associated risk factors in farms, and haplotypes isolated from the native village chicken and pig populations in Peninsular Malaysia. The seroprevalence of *T. gondii* in village chickens at the animal level was low at 7.6% (95% CI: 4.60–11.60), while at the farm level, it was 52.0% (95% CI: 31.30–72.20). For pigs, the animal-level seroprevalence of *T. gondii* was 3.0% (95% CI: 1.60–5.10), while the farm-level, it was 31.6% (95% CI: 12.60–56.60). The PCR-based DNA detection on meat samples from chickens (*n* = 250) and pork (*n* = 121) detected 14.0% (95% CI: 9.95–18.9) and 5.8% (95% CI: 2.4–11.6) positive, respectively. Six unique *T. gondii* haplotypes were isolated from the tissue samples. Multivariable logistic regression analysis showed that feeding the chickens farm-produced feeds and allowing wild animals access to pig farms were significant determinants for farm-level seropositivity. Providing hygienic and good quality feeds to chickens and increasing biosecurity in pig farms through prevention of access by wildlife may reduce the risk of transmission of *T. gondii* infection in the local chickens and pig farms.

## 1. Introduction

*Toxoplasma gondii* is a re-emerging zoonotic parasite capable of causing serious morbidity and mortality in almost all warm-blooded animal species worldwide [1]. One-third of the global human population is infected with toxoplasmosis [2], usually through ingestion of raw or undercooked meat from infected animals or water or soil contaminated with oocyst from the faeces of infected felids [3]. Toxoplasmosis is a life-threatening infection for the unborn foetus and immunocompromised individuals [4,5], manifesting various symptoms, such as seizures, nausea, and poor coordination in humans [6]. Recently, the apparent increase in toxoplasmosis foodborne infection in developed countries and concerns about the suggested link between mental illness and seropositivity to the agent have brought renewed interest in the protozoa [3,5].

Chicken and pork are two major protein sources in Malaysia [7]. Village chicken meat is perceived to be safer, possesses medicinal properties, and is more wholesome than commercial broilers, as little or no chemicals or drugs are routinely used in their production [8,9]. Moreover, the changing consumer preference and increase in the level of income have increased the demand for village chickens [10]. However, according to a few authors, village chickens may increase the risk of *T. gondii* infection in humans because these chickens are highly exposed to the oocysts in the environment [11,12]. On the other hand, pigs are an important reservoir of *T. gondii*, and pork is purported to be a significant source of infection for humans in different parts of the world [13,14]. Infected pigs are usually asymptomatic, and the infective cysts in their tissue are too small to be detected with the naked eye [13]. Infection in pigs is mainly acquired through ingestion of infected intermediate hosts, contaminated feed or water, cannibalism, and other vices, such as ear and tail biting [15]. The infection in pigs is most of the time self-limiting, but abortions and death may occur in severe cases [16].

This study aimed to determine the prevalence of *T. gondii* among village chicken and pig populations and to identify factors that may influence the presence of the pathogen in livestock farms in the study area. In addition, this study also discovers the haplotypes of *T. gondii* circulating in Malaysian village chickens and pigs. This may improve the understanding of the epidemiology of the agent in these species.

## 2. Materials and Methods

### 2.1. Study Areas

Malaysia is separated into Peninsular Malaysia in the west, and East Malaysia is located on Borneo Island, comprising the states of Sabah and Sarawak. These two parts occupy 329,847 km^2^ and 132,265 km^2^, respectively, and are separated by the South China Sea [17]. The peninsula comprises 12 states and the 2 administrative capitals of Kuala Lumpur and Putrajaya. This study was conducted in five states in Peninsular Malaysia, including Penang, Perak, Selangor, Melaka, and Johor (Figure 1).

### 2.2. Study Design and Sample Size Estimation

A cross-sectional study was conducted between February 2019 and September 2020. The sample size for the study was calculated using OpenEpi (version 2.3) statistical software, assuming an expected seroprevalence of 20% based on an earlier study by Sabri et al. [18] for village chickens and 50% for pig samples. With a desired absolute confidence level of 95%, an absolute accuracy of 10%, and a large population size, a total of at least 246 serum samples from adult village chickens and 384 finisher pigs were estimated to be collected in this study. The selection of states in Peninsular Malaysia for farm sampling was based on the largest population size of chickens and pigs. The sampling frame was obtained from data published by the Department of Veterinary Services Malaysia [19]. Random village chicken farms were selected from each of the four study sites (Perak, Selangor, Melaka, and Johor) using a random number generated by Microsoft Excel^®^ 2011. For pigs, the state DVS was contacted and permission for the study protocol was granted by the Research and Innovation Division of the DVS. Pig farms were selected by the DVS in the five different states (Penang, Perak, Selangor, Melaka, and Johor) based on their ongoing annual disease surveillance work. Selected farms were contacted via telephone to seek their consent for participation in the study. Subsequently, an appointment for a visit was set. Given the unknown farm-level prevalence and rarity of infections, the number of village chicken and pig farms to be included in this study was calculated using the assumption that the parasite can be detected in at least 10% of the farms with the precision of 10% at the confidence level of 95%. The total number of farms was calculated as 31 and 33 farms each for village chickens and pigs, respectively.

### 2.3. Farm Characteristics

For data collection, the management of the chicken farms is categorized as either free-range or intensive. A farm whose birds were left free to roam and scavenge during the day, but return to the shed at dusk, fed with leftovers and household scraps, were considered free-range farms [20]. On the other hand, farms whose birds are housed in a fenced outdoor area, permitted to roam outdoors within a limited space, and housing types vary from simple cages to more advanced housing were considered intensive farms [21].

The most common management system practised in pig farms is the intensive farm system, categorized into an open-house and close-house system [22]. An open-house farm is characterized by an open-sided house with natural ventilation, not properly gated, stray animals may access the pens, and the pigs are kept in confined pens with adequate shelter on a concrete or hard floor with sufficient feeding and watering facilities. On the other hand, the closed-house system is characterized as Modern Pig Farming (MPF) and abides by the strict regulatory policies of the government to prevent environmental pollution. They have proper ventilation with cooling pads, zero discharge of wastewater into the public drainage, and measures were put in place to prevent stray animals from accessing the farm [23,24].

The inclusion criteria for the village chicken farms are that farms that consented to participate and had a minimum of 10 birds aged 6–12 months were enrolled in the study. For pig farms, any farm with a minimum of 25 pigs aged more than five months and consented to participate was enrolled.

### 2.4. Blood Sample Collection

Prior to the commencement of the study, the protocol, which involved the slaughter of chickens and collection of blood from live pigs, was approved by the Animal Care and Use Committee (ACUC) of Universiti Putra Malaysia vide Animal Use Protocol (AUP) reference number UPM/IACUC/AUP-R033/2019. To collect samples from chickens, a batch of 10 chickens was purchased from each farm, kept in a cage, and provided feed and water ad libitum. The batch from each farm was slaughtered on-site, and 3 mL of blood was collected from each chicken. For pigs, finisher pigs were housed in shaded stalls and provided feed and water ad libitum. A batch of 25 pigs was handled per session of sample collection. Each pig was restrained, and 3 mL of blood was collected from the jugular vein using an 18-gauge needle attached to a vacutainer tube. The tubes were labelled using farm and animal identification codes. Samples from farms in Selangor state were transported on ice packs to the Parasitology Laboratory, Faculty of Veterinary Medicine, Universiti Putra Malaysia, while samples from other states were transported to the regional veterinary laboratories of the DVS for processing.

### 2.5. Blood Sample Processing and Serological Examination

The samples from chickens and pigs were centrifuged at 4000 g for 10 min in a refrigerated centrifuge (Eppendorf, Centrifuge 5804R, Hamburg, Germany). The sera were then transferred into a 1.5 mL microcentrifuge tube using micropipettes and stored at −20 °C until use. Sera samples from chickens were tested for antibodies to *T. gondii* using animal species-specific indirect commercial ELISA kits. *Toxoplasma* Circulating Antigen TCA ELISA kit (Sunlong Biotechnology, Shanghai, China) and Porcine Toxoplasmosis Antibodies ELISA kit (Elabscience Biotechnology Co., Ltd., Houston, TX, USA) were used for the chicken and pig samples, respectively. Manufacturers of the two kits reported sensitivity and specificity values of more than 98.0%. The ELISA test procedures were carried out according to the manufacturer’s instructions.

### 2.6. Tissue Sample Collection

A total of 250 tissue samples were collected from the slaughtered chickens during farm visits. About 100 g of samples, including brain, heart, lung, and pectoralis muscle tissues, were collected from each chicken. Tissues from the same birth were pooled, sealed in clean plastic bags, and labelled using an identification code. Thereafter, the samples were placed on ice packs and transported to the laboratory and stored at −20 °C until use.

To determine the proportion of *T. gondii* infection in pigs, a total of 121 pork samples were purchased based on availability in butchers’ shops and meat kiosks located in pig farms. The tissue samples collected included tongue (*n* = 36), diaphragm (*n* = 17), and intercostal muscle (*n* = 68). The approximate weights of the samples were between 50 g and 100 g. Among these, 91 pork samples were purchased from 8 different butcher shops in Selangor, 21 samples were collected from three farms in Selangor, and 9 samples were collected from 2 farms in Penang. The farms from which tissue samples were collected were different from those from which blood samples for the serological investigation were collected. Each of the pork tissue samples was analysed separately, assuming each originated from a different animal. Each sample was kept in a properly labelled plastic bag and preserved in a −20 °C refrigerator pending further processing.

### 2.7. Sample Processing and DNA Extraction

The frozen tissue samples were thawed, excess fats were trimmed, and the fat-free tissue samples were minced. To avoid contamination between samples, the instruments were cleaned and decontaminated between each sample. The samples were mixed with 10 mL of phosphate-buffered saline buffer (PBS pH 7.4, Thermo Fisher, Waltham, MA, USA) in a stomacher bag and homogenised using a stomacher (BagMixer Interscience, St. Nom la Breteche, Sainte-Non-labour-Taise, France) for 2 min. Then, 200 µL of the liquid content of the bag was transferred into a 2 mL microcentrifuge tube for DNA extraction. The DNA extraction from samples was conducted using Geneaid Genomic Tissue DNA Kit (Geneaid Biotech Ltd., Taiwan) based on the manufacturer’s instructions.

### 2.8. PCR Amplification, Cloning and Sequencing

Nested-PCR was performed using the internal transcribed spacer 1 (ITS1) region primers targeting a 227 bp fragment of the ITS1 region as previously described by Hurtado et al. [25]. Two primer sets were used: external primer set NN1 (CCTTTGAATCCCAAGCAAAACATGAG) and NN2 (GCGAGCCAAGACATCCA TTGCTGA) for primary amplification, and internal primer set Tg-NP1 (GTGATAGTATCGAAAGGTAT) and Tg-NP2 (ACTCTCTCTCAAATGTTCCT) for secondary amplification. The 25 μL reaction mixtures for the first PCR contained 12 μL of 2x TopTaq Master Mix Kit (Qiagen, Hilden, Germany), 1 μL of 0.2 μM of each primer, 5 μL of DNA template, and 6 μL of ddH_2_O. The mixture was subjected to the following cycling conditions: initial heating step at 94 °C for 3 min, followed by 15 cycles of denaturation at 94 °C for 30 s, annealing at 65 °C for 45 s, extension at 72 °C for 1 min, and final extension at 72 °C for 5 min. In the secondary amplification, 2.5 μL from the PCR product of the first amplification was added to the tubes containing new PCR reaction mixtures. The initial reaction was set at 94 °C for 3 min, followed by 35 cycles of denaturing at 94 °C for 20 s, annealing at 53 °C for 30 s, extension at 72 °C for 30 s, and final extension step at 72 °C for 5 min. PCR positive control (confirmed *T. gondii* DNA extracted from wild rats) and negative control (RNase-free water) were included in each PCR run. The PCR products were resolved in a 1.5% agarose gel by electrophoresis and imaged by Gel Doc XR Plus (Bio-Rad, Hercules, CA, USA). A 100 bp DNA ladder (Qiagen, GmbH, Germany) was used as a marker.

The QIAquick Gel Extraction Kit (Qiagen, Germany) and CloneJET PCR Cloning Kit (Thermo Fisher, USA) were used to extract and clone the positive PCR amplicons according to the manufacturer’s instructions, respectively. The transformed cells were then plated evenly on pre-warmed Luria-Bertani (LB) with an ampicillin agar plate and incubated overnight at 37 °C. Ten colonies from each plate were screened by direct colony PCR using specific primers described previously to check for gene insertion. Positive amplicons were sent for sequencing and compared to known generic sequences of the ITS1 region curated by the National Centre for Biotechnology Information (NCBI) GenBank using the BLAST tool.

### 2.9. Data Collection

To collect information on the characteristics of selected farms, a closed-ended questionnaire was developed. An initial pilot study was conducted involving a few livestock farms to appraise the understanding of the questions in the instrument, identify potential problems, and estimate the time required for the completion of the questions. The list of items in the initial questionnaire pertaining to potential farm-level risk factors for *T. gondii* infection in livestock farms was extracted from the published works of Herrero et al. [26] and Stelzer et al. [27]. Based on the results from the pre-test, the initial draft questionnaire was modified to the final version used for the data collection. The items in the questionnaire were structured to collect information on farm demography, farm management and biosecurity measures at the farm. The detailed questionnaire was included as Appendix A.

### 2.10. Data Analysis

The serological and molecular prevalence of *T. gondii* positive for each animal species were determined by dividing the number tested positive by the total number tested, and the results were presented as percentages. Prevalence at the animal and farm levels was calculated with their respective 95% confidence intervals (95% CI). A farm is considered positive if at least one animal tests positive. Serological results were cross tabulated with the potential risk factor using the Pearson Chi-square test (or Fisher’s exact test) and were considered statistically significant when *α* ≤ 0.05. Multilevel logistics regression was considered for further analysis of the risk factors due to the potential clustering of data within a farm and was carried out as recommended by Crowson [28]. However, the intra-cluster correlation (ICC) was found to be *ρ* < 0, reflecting poor reliability and suggesting that a multilevel logistic regression was not suitable for the data analysis. Therefore, risk factors with *α* ≤ 0.2 in the univariate analysis were selected [29] and analysed by multivariable logistic regression model using the forward Wald method. The problems of empty cells in tables due to small sample was addressed using Haldane Correction as described by Greenland et al. [30] for the calculation of odds ratios. All statistical analyses were performed using IBM SPSS Statistics version 28 (IBM, Armonk, NY, USA: IBM Corp.) at the significance level α = 0.05.

For phylogenetic analysis, the DNA sequence electropherograms obtained from the clones were manually checked using BioEdit v7.0.9 [31] to resolve ambiguity. They were then subjected to multiple alignments in ClustalX [32] using default parameters to obtain consensus sequences. A sequence was considered a unique haplotype if at least two clones contained the putative haplotype. Phylogenetic analysis of the nucleotide sequences was performed using MEGA11. All positions containing gaps and missing data were eliminated. Maximum Likelihood (ML) and Neighbour-Joining (NJ) trees were constructed to determine the phylogenetic affinities.

## 3. Results

### 3.1. Descriptive Statistics of Farms

A total of 250 sera samples were collected from 25 village chicken farms and 433 sera from 19 pig farms. The highest number of chicken samples was obtained from Selangor, while Penang and Melaka were for pigs. The majority of the chicken farms visited in the study practised a free-range farming system (88%), while most of the pig farms practiced an open-house farming system (89.47%).

### 3.2. Seroprevalence and Risk Factors of Toxoplasma Gondii

Toxoplasmosis is widely spread amongst tested farms (>30%), but comparatively low numbers of animals per farm were infected (<10%) (Table 1). In the univariable analysis for risk factors, the ‘type of feed’ given to village chickens (Table 2) and ‘wild animals having possible contact with pigs’ were significantly associated (*p* < 0.05) with farm-level seropositivity (Table 3). Multivariable logistics found ‘produced feed on farm’ increases the likelihood of chicken farm-level seropositive by 6.8 times (95% CI: 1.2–29.2), while contact with wild animals, such as rats, increased the odds of seropositivity in pig farms by 16.7 times (95% CI: 1.2–204.0).

### 3.3. Molecular Prevalence, Haplotypes and Phylogenetic Tree of Toxoplasma gondii

*T. gondii* DNA was detected in 14% (95% CI: 9.95–18.93) of village chicken tissue samples. For pigs, a molecular prevalence of 5.79% (95% CI: 2.36–11.56) was detected in various organs. Test results based on organs showed that four (57%), two (29%), and one (14%) were positive for *T. gondii* DNA in the tongue, intercostal muscle, and diaphragm of pork, respectively. Six unique haplotypes (TgMH01-06) were isolated from tissue samples from village chicken (100%) and pork (33.33%), with TgMH01 (19.47%) and TgMH05 (25.66%) recording the highest number of haplotype frequency. These sequences bear GenBank accession numbers OP490598–OP490603. No spatial clustering of *T. gondii* haplotypes was observed in the isolates in this study.

## 4. Discussion

Chickens and pigs are the most significant reservoirs for *T. gondii* transmission and, therefore, a public health and food safety concern. Previous toxoplasmosis surveys on cattle, pigs, sheep, and cats in Malaysia reported seroprevalence ranging between 0–35.5% [33,34,35,36]. In the present study, the seroprevalence of *T. gondii* in village chickens and pigs was similar at 7.6% and 3%, respectively. Despite the relatively low seroprevalence of *T. gondii* in the animals in this study, the farm-level seroprevalence was high in both animal species. The higher farm-level seroprevalence reflects the wide distribution of the oocysts that are able to remain viable in the environment for a long time [1,37].

Modifications in pig production and management have resulted in a lower seroprevalence of *T. gondii* in Malaysian pigs [33,38]. While the 3% animal-level seroprevalence of toxoplasmosis in pigs we are reporting here corroborates the report of intensively raised pigs from Indonesia (2.3%) [39], a higher seroprevalence was found in indoor-reared pigs from Denmark (33.7%) [40]. On the other hand, Spain (85%) reported higher farm-level seroprevalence in pigs from the intensive farms [41], but similar to the findings from a similar population in Greece (26.2%) [15]. Danish sows (over one year of age) recorded higher prevalence due to the accumulative exposure to *Toxoplasma*, thereby suggesting older animals are more prone to the infection compared to pigs with less than one year of age.

Chickens provided with homemade feed have an increased chance of ingesting viable *T. gondii* tissue cysts in the food scraps. Our finding agrees with previous studies in village chickens from Brazil [42,43], where toxoplasmosis increased when chickens were fed with food scraps made from leftover human food, vegetables, and raw animal viscera. Moreover, stray cats are attracted to household leftovers and can excrete millions of oocysts on farms.

The close contact between wild animals and pigs was a notable source of *T. gondii* infection. Moreover, in the surveyed areas, open-sided pig stables are common and are not completely enclosed from the outside environment, thus permitted contact between wildlife and pigs. Wildlife harbours the parasite as tissue cysts and are mechanical transmitters of *T. gondii* oocysts [44,45]. Pigs are omnivores and are known to eat rodents or their cadavers. Furthermore, *T. gondii* causes changes in rodent behaviour (e.g., causing neurological impairment) that reduce their innate fear of predators [46], thus increasing the risk of predation by pigs. Birds can also contribute to the increased risk of toxoplasmosis in pigs by spreading the parasite across wide geographical areas. Therefore, good biosecurity practices at the animal farm and insurance that these animals remain segregated from wildlife should be emphasised.

The molecular prevalence findings of chickens in our study were lower than free-roaming chickens from the West Indies (41%) [47], but are similar to a report in backyard chickens from Brazil (16.7%) [48]. For pigs, the molecular prevalence in this study corroborates with that reported in intensively reared pigs from Spain (8%) [49] but was lower than a study in conventionally raised pigs from Serbia (20.4%) [50]. Studies reported *T. gondii* DNA is found higher in the brains and hearts than in other tissues [51,52,53]. In our study, most of those organs were not available at the butcher shop and market sampled and therefore were not included in the analyses.

Phylogenetic analysis of *T. gondii* sequences from this study was clustered together with isolates from other studies (Figure 2 and Figure 3). This indicates close genetic relatedness of the isolates, which corroborates previous reports on chickens from the USA [54] and cats from China [55]. According to Bontell et al. [56], sexual recombinants of *T. gondii* in cats are rare in nature, as transmissions commonly occur through carnivorism and scavenging. Moreover, it has been suggested that the greater genetic diversity of the parasite in the wild compared to domestic settings is due to higher host diversity [57]. Our finding suggests the rarity of sexual recombinants of *T. gondii* in domestic village chickens and pigs, where transmission is mainly through ingesting contaminated food and infected intermediate hosts, thus leading to limited genetic variants of *T. gondii* recovered.

### Limitations of the Study

This study has limitations that need to be taken into consideration, such as the samples were taken from the chickens and pigs at their respective market age and, therefore, may not be representative of all the segments of the chicken and pig populations in the study area. In addition, even though the minimum sample size for the number of individual chickens and pigs for the study was achieved, that for the number of farms was not due to restrictions during the pandemic and improved biosecurity to prevent emerging transboundary diseases. Therefore, the risk factor analysis may not have enough power to detect differences in the proportions, resulting in wide confidence intervals for the odds ratio estimates.

## 5. Conclusions

*T. gondii* is present in village chickens and pigs with limited genetic variability in Peninsular Malaysia. The serological and molecular prevalence of the agent at the animal level was relatively lower compared to values reported elsewhere. However, the farm-level prevalence was high for both animal species. Feeding farm-made feeds to village chickens and allowing wild animals access to pig farms may increase the risk of exposure of the pigs to *T. gondii* infection. Therefore, avoiding these risk factors by providing hygienic and properly cooked feedstuff to chickens and increasing biosecurity measures in pig farms should be implemented as intervention strategies.

## Figures and Tables

**Figure 1 vetsci-10-00334-f001:**
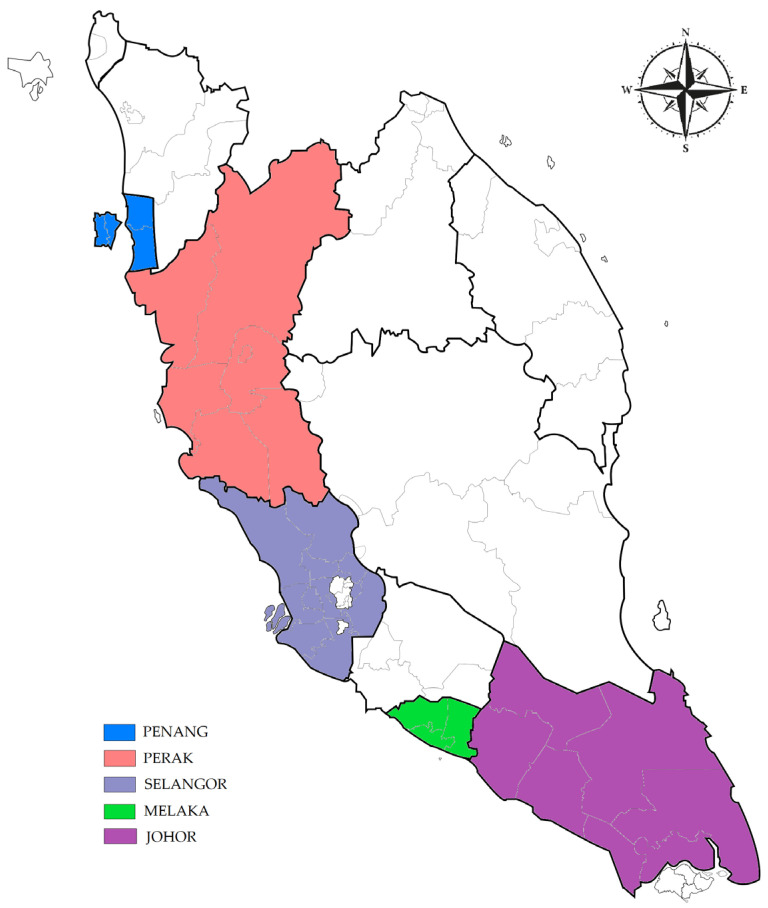
A map of Peninsular Malaysia showing the study areas.

**Figure 2 vetsci-10-00334-f002:**
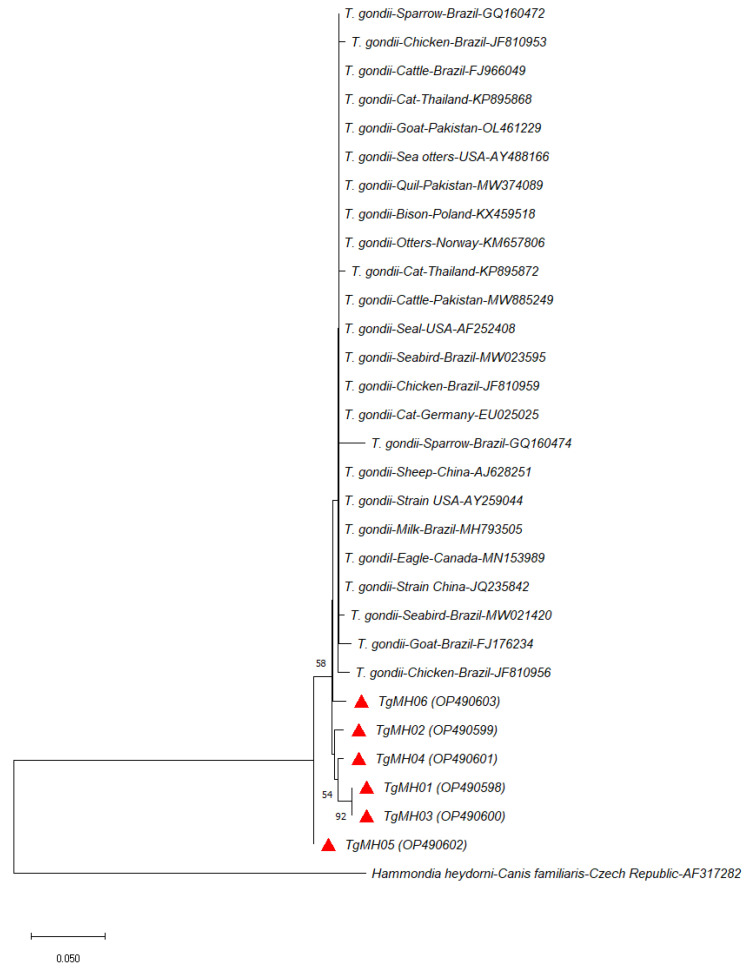
Neighbour-Joining phylogenetic tree (1000 bootstrap replicates) of *Toxoplasma gondii* based on 224 nucleotide residues of the ITS1 region computed using the *p*-distance model. The *T. gondii* haplotypes isolated from village chickens and pigs in this study are marked as red triangles with NCBI GenBank accession numbers in parenthesis. *Hammondia heydorni* (AF317282) was included as an outgroup.

**Figure 3 vetsci-10-00334-f003:**
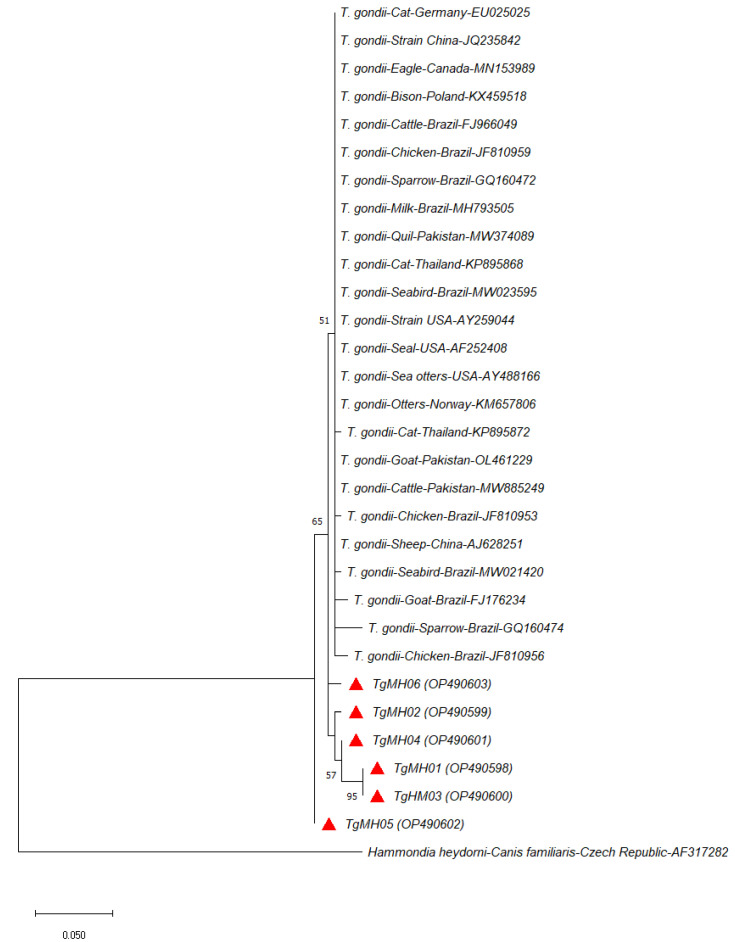
Maximum Likelihood phylogenetic tree (100 bootstrap replicates) of *Toxoplasma gondii* based on 224 nucleotide residues of the ITS1 region computed using the Jukes Cantor model. The *T. gondii* haplotypes isolated from village chickens and pigs in this study are marked as red triangles with NCBI GenBank accession numbers in parenthesis. *Hammondia heydorni* (AF317282) was included as an outgroup.

**Table 1 vetsci-10-00334-t001:** Seroprevalence of *Toxoplasma gondii* in village chickens and pigs from selected states in Peninsular Malaysia.

Variable	Categories	No. Animals Tested	Positive (%)	95% CI	No. Farms Tested	Positive (%)	95% CI
Chickens	-	250	19 (7.60)	4.60–11.60	25	13 (52.00)	31.30–72.20
State	Perak	50	3 (6.00)	1.26–16.55	5	2 (40.00)	5.27–85.34
	Selangor	90	8 (8.89)	3.92–16.76	9	4 (44.40)	13.70–78.80
	Melaka	60	5 (8.33)	2.76–18.39	6	4 (66.67)	22.28–95.67
	Johor	50	3 (6.00)	1.26–16.55	5	3 (60.00)	14.66–94.73
Pigs	-	433	13 (3.00)	1.60–5.10	19	6 (31.58)	12.60–56.60
State	Penang	102	7 (6.86)	2.80–13.63	4	3 (75.00)	19.41–99.37
	Perak	100	0	-	4	0	-
	Selangor	28	0	-	3	0	-
	Melaka	102	2 (1.96)	1.00–6.90	4	2 (50.00)	6.76–93.24
	Johor	101	4 (3.96)	1.09–9.83	4	1 (25.00)	1.00–80.59

**Table 2 vetsci-10-00334-t002:** Univariable analysis of risk factors in farms associated with seropositivity for *Toxoplasma gondii* among village chickens in Peninsular Malaysia.

Risk Factor	Categories	Frequency	Positive (%)	Chi-Square (χ^2^)	Odds Ratio	*p*-Value
States	Perak	5	2 (40.00)	0.40	0.44 (0.04–5.58)	1.00
	Selangor	9	4 (44.44)	0.31	0.53 (0.06–4.91)	1.00
	Melaka	6	4 (66.67)	0.05	1.33 (0.11–15.70)	1.00
	Johor	5	3 (60.00)		1.00 ^a^	
Farm system	Free-range	22	12 (54.55)	0.48	2.40 (0.19–30.52)	0.59
	Caged	3	1 (33.33)		1.00 ^a^	
Type of feed	Produced on farm	12	9 (75.00)	4.89	6.75 (1.16–39.20)	0.03
	Commercial	13	4 (30.77)		1.00 ^a^	
Feed storage	Close	22	10 (45.45)	1.96	0.14 (0.01–3.11)	0.16
	Open	3	3 (100)		1.00 ^a^	
Feed location	On ground	1	1 (100.00)	0.16	2.00 (0.06–65.41)	1.00
	Off ground	24	12 (50.00)		1.00 ^a^	
Water source	Pipe	20	10 (50.00)	0.16	0.67 (0.09–4.89)	1.00
	Well	5	3 (60.00)		1.00 ^a^	
Farm often clean	Yes	12	8 (66.67)	1.99	3.20 (0.62–16.49)	0.16
	No	13	5 (38.46)		1.00 ^a^	
Presence of other livestock	Yes	13	9 (69.23)	3.22	4.50 (0.84–24.18)	0.07
	No	12	4 (33.33)		1.00 ^a^	
Presence of cat/dog	Yes	20	12 (60.00)	2.56	6.00 (0.56–63.98)	0.16
	No	5	1 (20.00)		1.00 ^a^	
Presence of rodents	Yes	24	12 (50.00)	0.16	0.5 (0.01–16.35)	0.71
	No	1	1 (100.00)		1.00 ^a^	
Contact with wild animals	Yes	8	6 (75.00)	2.49	4.29 (0.66–27.79)	0.20
	No	17	7 (41.18)		1.00 ^a^	
Other animals access feed	Yes	7	5 (71.43)	1.47	3.13 (0.48–20.58)	0.38
	No	18	8 (44.44)		1.00 ^a^	
Rodent control program	Yes	24	12 (50.00)	0.96	0.5 (0.01–16.35)	0.71
	No	1	1 (100.00)		1.00 ^a^	

^a^ Reference category.

**Table 3 vetsci-10-00334-t003:** Univariable analysis for risk factors in farms associated with seropositivity for *Toxoplasma gondii* among pigs in Peninsular Malaysia.

Risk Factor	Categories	Frequency	Positive (%)	Chi-Square (χ^2^)	Odds Ratio	*p*-Value
States	Penang	4	3 (75.00)	2.00	9.00 (0.37–220.90)	0.49
	Perak	4	0	0.21	0.43 (0.01–17.83)	1.00
	Selangor	3	0	0.07	0.60 (0.01–26.47)	1.00
	Melaka	4	2 (50.00)	0.03	3.00 (0.15–59.88)	1.00
	Johor	4	1 (25.00)		1.00 ^a^	
Farm system	Close	2	0	0.23	0.46 (0.02–11.87)	1.00
	Open	17	6 (35.29)		1.00 ^a^	
Type of feed	Produced on farm	6	2 (33.33)	0.01	1.13 (0.14–8.88)	1.00
	Commercial	13	4 (30.77)		1.00 ^a^	
Feed storage	Open	6	2 (33.33)	0.01	1.13 (0.14–8.88)	1.00
	Close	13	3 (100)		1.00 ^a^	
Feed location	On ground	3	1 (33.33)	0.01	1.10 (0.08–15.15)	1.00
	Off ground	16	5 (31.25)		1.00 ^a^	
Water source	Pipe	3	2 (66.67)	2.03	6.00 (0.42–85.25)	0.20
	Well	16	4 (25.00)		1.00 ^a^	
Farm often clean	Yes	17	5 (29.41)	0.35	0.42 (0.02–8.05)	1.00
	No	2	1 (28.57)		1.00 ^a^	
Presence of other livestock	Yes	5	2 (40.00)	0.22	1.67 (0.20–14.05)	1.00
	No	14	4 (28.57)		1.00 ^a^	
Presence of cat/dog	Yes	17	6 (60.00)	0.23	2.18 (0.08–56.52)	1.00
	No	2	0		1.00 ^a^	
Presence of rodents	Yes	14	5 (35.71)	0.18	0.43 (0.01–24.15)	1.00
	No	5	1 (20.00)		1.00 ^a^	
Contact with wild animals	Yes	8	5 (62.50)	6.12	16.67 (1.36–204)	0.04
	No	11	1 (9.09)		1.00 ^a^	
Other animals access feed	Yes	5	3 (60.00)	2.54	5.50 (0.61–49.54)	0.26
	No	14	3 (21.43)		1.00 ^a^	
Rodent control program	Yes	18	6 (33.33)	0.49	0.5 (0.01–28.42)	0.68
	No	1	0		1.00 ^a^	

^a^ Reference category.

## Data Availability

The datasets utilized for this study are available from the corresponding author on request.

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
