# Peer review of "Prevalence and Haplotypes of Toxoplasma gondii in Native Village Chickens and Pigs in Peninsular Malaysia"

_vetsci, 2023, doi:10.3390/vetsci10050334_

Round 1
Reviewer 1 Report
The discussion is weak, not bringing relevant contributions that justify the work. Lots of speculation. Most of the discussion is based on other articles, without the present work being directed to allow the presented conclusions. The limitations of the study show that there was no strict criterion for selecting the samples, which conflicts with several results from other authors. Some of the issues raised in the discussion should have been considered during experimental design.
Reviewer 2 Report
This is a very importante type of study. Toxoplasmosis as a foodborne infection is becoming more and more important. Infection through the ingestion of raw or undercooked contaminated meat is associated with several outbreaks. Material and Methods are clear. However, a map with the geopraphic location of the samples would help and should be included.
These study has two major limitations:
First - no strain was isolated. The isolates were based on DNA detection
Second - Line 56 – “this study also discovers the haplotype diversity of T. gondii...”. In fact, 6 haplotypes were defined, but only based on the sequencing of a single marker. This is not correct. This gave us a limited information. A multi-locus approach should be followed for a more accurate genotyping and more discriminative power.
To a correct definition of the haplotypes, the authors should included the study of more markers.
Reviewer 3 Report
This is a very interesting article, on a protozoan which is common, causes major issues among humans and animals, but is rarely reported in this sense. So it is very nice to see a study of this standard filling that gap. The study appears well organised, well designed, and well written, and as a result, I only have a few comments which are mainly grammatical which I have detailed below
Line 114- 6-12 months which were enrolled in the study …. (reword)
Line 124 and 126- ad libitum should be italicised
Line 169- would between each sample sound better than between one sample and the other? Up to the authors
Line 170- please add a manufacturer for the PBS.
Line 193- its quite unusual to run the ladder through the PCR assay- is there a reason why this was done?
Line 203- Genbank rather than Genebank
Figure 1 and 2, I would question the root choice here, as it limits the visibility of the Toxoplasma variation. Is it possible to choose a more closely related organism? Or even cut the branches of the tree a bit to show the Toxoplasma variation more?
Line 341- sows (over one year of age) …. (reword)
Line 362- environment permitted contact between ….. (reword)
Line 392- results of chickens was lower than …. (reword)
But well done to the authors on an interesting and well written study.
Round 2
Reviewer 1 Report
We believe that the relevant changes, especially regarding the discussion, were satisfactory and that they improved the quality of the proposed work. The article in its current format is ready to be published.
Reviewer 2 Report
Dear Authors, I have no more questions